# Phosphorylated Peptide Derived from the Myosin Phosphatase Target Subunit Is a Novel Inhibitor of Protein Phosphatase-1

**DOI:** 10.3390/ijms24054789

**Published:** 2023-03-01

**Authors:** Zoltán Kónya, István Tamás, Bálint Bécsi, Beáta Lontay, Mária Raics, István Timári, Katalin E. Kövér, Ferenc Erdődi

**Affiliations:** 1Department of Medical Chemistry, Faculty of Medicine, University of Debrecen, H-4032 Debrecen, Hungary; 2Department of Inorganic and Analytical Chemistry, Faculty of Natural Science and Technology, University of Debrecen, H-4032 Debrecen, Hungary; 3Department of Organic Chemistry, Faculty of Natural Science and Technology, University of Debrecen, H-4032 Debrecen, Hungary; 4MTA-DE Molecular Recognition and Interaction Research Group, University of Debrecen, H-4032 Debrecen, Hungary

**Keywords:** protein phosphatase-1 (PP1), myosin phosphatase (MP), myosin phosphatase target subunit-1 (MYPT1), MYPT1 inhibitory peptide (P-Thr696-MYPT1^690−701^), molecular docking, saturation transfer difference NMR

## Abstract

Identification of specific protein phosphatase-1 (PP1) inhibitors is of special importance regarding the study of its cellular functions and may have therapeutic values in diseases coupled to signaling processes. In this study, we prove that a phosphorylated peptide of the inhibitory region of myosin phosphatase (MP) target subunit (MYPT1), R^690^QSRRS(pT696)QGVTL^701^ (P-Thr696-MYPT1^690−701^), interacts with and inhibits the PP1 catalytic subunit (PP1c, IC_50_ = 3.84 µM) and the MP holoenzyme (Flag-MYPT1-PP1c, IC_50_ = 3.84 µM). Saturation transfer difference NMR measurements established binding of hydrophobic and basic regions of P-Thr696-MYPT1^690−701^ to PP1c, suggesting interactions with the hydrophobic and acidic substrate binding grooves. P-Thr696-MYPT1^690−701^ was dephosphorylated by PP1c slowly (t_1/2_ = 81.6–87.9 min), which was further impeded (t_1/2_ = 103 min) in the presence of the phosphorylated 20 kDa myosin light chain (P-MLC20). In contrast, P-Thr696-MYPT1^690−701^ (10–500 µM) slowed down the dephosphorylation of P-MLC20 (t_1/2_ = 1.69 min) significantly (t_1/2_ = 2.49–10.06 min). These data are compatible with an unfair competition mechanism between the inhibitory phosphopeptide and the phosphosubstrate. Docking simulations of the PP1c-P-MYPT1^690−701^ complexes with phosphothreonine (PP1c-P-Thr696-MYPT1^690−701^) or phosphoserine (PP1c-P-Ser696-MYPT1^690−701^) suggested their distinct poses on the surface of PP1c. In addition, the arrangements and distances of the surrounding coordinating residues of PP1c around the phosphothreonine or phosphoserine at the active site were distinct, which may account for their different hydrolysis rate. It is presumed that P-Thr696-MYPT1^690−701^ binds tightly at the active center but the phosphoester hydrolysis is less preferable compared to P-Ser696-MYPT1^690−701^ or phosphoserine substrates. Moreover, the inhibitory phosphopeptide may serve as a template to synthesize cell permeable PP1-specific peptide inhibitors.

## 1. Introduction

Phosphorylation of proteins on serine/threonine (Ser/Thr) residues is an important posttranslational regulatory mechanism in many cellular processes [1]. The phosphorylation level of proteins at distinct physiological challenges reflects the balance of the activities of the phosphorylating protein kinases and the dephosphorylating protein phosphatases. A plethora of protein kinases and phosphatases accomplish these regulatory functions in cells; therefore, identification of the specific enzymes involved in distinct signaling pathways is of special importance [2]. Moreover, finding effectors of possible pharmacological relevance to influence the activities of these interconverting enzymes is also an important issue. To this end, protein kinases received more attention than protein phosphatases at the beginning of the “phosphorylation era”; however, this trend has been changed during the past few decades, as phosphatases have also been considered as “druggable” targets [3].

Protein phosphatase-1 (PP1) is a major phospho-Ser/Thr (P-Ser/Thr)-specific enzyme that plays important roles in the regulation of many metabolic and other cellular processes [4]. PP1 occurs in cells in holoenzyme forms in which one of the PP1 catalytic subunit (PP1c) isoforms (PP1cα, PP1cβ/δ or PP1cγ) is associated with regulatory subunits and/or inhibitory protein(s) [5]. The 3D structure of PP1c in complex with tungstate [6], inhibitory toxins [7,8,9] or regulatory subunit peptides [10,11] were determined. These studies identified the major surfaces for binding substrates [7] and regulatory proteins [10,11] as well as describing the structure of the catalytic center where the hydrolysis of the phosphoester occurs with the help of two metal ions coordinated by several PP1c amino acid side chains [6,7]. PP1c includes three substrate-binding surfaces: the hydrophobic, acidic and C-terminal grooves, which form a Y-shape with the active site at the bifurcation point. Inhibitory toxins (microcystin, okadaic acid) exert their suppressive effect on PP1 activity via interacting with the hydrophobic groove and part of the catalytic center [7,8] counteracting with substrate binding and hydrolysis. Physiological inhibitor proteins, such as inhibitor-1, dopamine and cAMP-regulated 32 kDa phosphoprotein (DARPP-32) and C-kinase activated phosphatase inhibitor of 17 kDa (CPI17), require phosphorylation for inhibition and they include hydrophobic and basic amino acids stretches N- and C-terminal to a phosphothreonine similar to as in some of the substrates, thereby blocking the substrate binding hydrophobic and acidic grooves as well as the active site of PP1c [11,12]. It has been established that phospho-CPI17 is also a substrate for PP1, but its dephosphorylation proceeds slower than that of the physiological phosphosubstrates; therefore, the inhibition is due to an “unfair competition” between the inhibitor and the substrate [13].

PP1c isoforms have limited substrate specificities; however, under cellular conditions PP1c regulatory/interacting proteins (PIPs) may ensure specific substrate recognition by targeting PP1c to different directions [5,14]. The interacting proteins bind to PP1 surfaces via a few short linear sequences of which the general (R/K)(V/I)X(F/W) motif (defined briefly as RVxF) is the best characterized with respect to the structure of PP1c–PIP complexes [10,11]. The binding surface in PP1c for RVxF motifs is distant from the catalytic center and from the substrate binding grooves. Stringent sequence search identified more than one hundred possible PIPs among mammalian proteins. These PIPs could be phosphosubstrates themselves, or they may function via specific domains targeting PP1c toward the phosphosubstrates, subcellular compartments or macromolecular complexes [15].

Inhibition or activation of protein phosphatases often result in opposite physiological outcomes: for examples, these influences may decrease [16] or increase [17] chemosensitivity of leukemic cells as shown by our earlier experiments. Compounds of natural origins such as several polyphenols [18] and ellagitannins [19] were also identified as potent inhibitors of PP1, but they were less effective on protein phosphatase-2A (PP2A). On the other hand, synthesized selenoglycosides in acetylated forms proved to be effective activators of both PP1c and PP2Ac [20]. These data are promising with respect to finding further possible physiological effectors of PP1. With these regards synthesized peptides including the RVxF motif, and several RVxFs mimic molecules have been shown to activate PP1 [21,22]. Inhibitory phosphopeptides, however, have not been probed as possible pharmacological effectors of PP1c.

Dissection of phosphorylated peptide regions from the inhibitory proteins (i.e., DARP-32 or inhibitor-1) may result in less effective inhibitors compared to the original proteins themselves [23]. Inhibitory phosphorylated regions are also present in regulatory subunits of PP1c, and myosin phosphatase (MP) holoenzyme (Figure 1) is an appropriate model for demonstrating the possible ways of the interactions and the regulatory role of the phosphorylation of the targeting subunit in the mediation of the phosphatase activity [24,25].

MP consists of PP1cδ, myosin phosphatase target subunit-1 (MYPT1) and a 20 kDa protein (M20) of little knowledge of its function. M20 binds to the C-terminal region of MYPT1, and it has no influence on the phosphatase activity. Moreover, M20 is present in distinct smooth muscles, but it is not detected in brain or skeletal muscle [24], for example, and its presence in different cell lines is not established either. These are the reasons why MP holoenzyme is often considered as a complex of PP1c and MYPT1 only in most of the studies as the presence of the M20 subunit is not revealed. In MYPT1, the RVxF motif is positioned at the N-terminal part (K^35^VKF^38^) and its presence is essential for interaction with PP1c. However, binding of the RVxF motif exposes further binding regions in MYPT1 in the N-terminal (MyPhone)-, ankyrin repeat- and acidic regions [26]. In addition, phosphorylation dependent binding sites for the 20 kDa myosin light chain (MLC20) and for the myosin heavy chains were also identified at the N-terminal ankyrin repeats [27] and at the C-terminal region [28], respectively. The C-terminal half of dephosphorylated MYPT1 does not interact with PP1c in the active MP holoenzyme. However, phosphorylation of MYPT1 at Thr696 (and at Thr853) by Rho-kinase in the C-terminal half of the protein results in inhibition of the activity of MP [29]. It is assumed that this inhibition is due to the interaction of the phosphorylated peptide region of MYPT1 (see Figure 1) with the substrate-binding grooves and the catalytic center of PP1c. Phosphorylated protein fragments of the C-terminal regions of MYPT1 have been characterized as possible PP1c [30] and MP [31] inhibitors before with conflicting results, but short phosphorylated peptides synthesized based on inhibitory phosphorylation sequences have not been investigated yet. The phosphorylated peptide (P-Thr696-MYPT1^690−701^) from MYPT1 (sequence is highlighted in Figure 1) appears to be a good candidate for testing as a PP1 inhibitor.

Our present study shows that P-Thr696-MYPT1^690−701^ interacts with and inhibits PP1c and the MP holoenzyme with similar effectiveness at micromolar concentration. It is shown that P-Thr696-MYPT1^690−701^ is a substrate for PP1, but its dephosphorylation proceeds much slower than the phosphorylated 20 kDa myosin light chain (P-MLC20) substrate suggesting an “unfair competition mechanism” for the inhibition established earlier [13]. 

## 2. Results

### 2.1. Inhibition and Interaction of PP1 with P-Thr696-MYPT1^690−701^ Peptide

We probed synthesized MYPT1 peptide—non-phosphorylated (MYPT1^690−701^) or phosphorylated—at Thr696 (P-Thr696-MYPT1^690−701^) on the activity of PP1c, PP2Ac and Flag-MYPT1-PP1c complex (Figure 2A). MYPT1^690−701^ was without effect while P-Thr696-MYPT1^690−701^ inhibited the activity of PP1c in a concentration dependent manner with an IC_50_ value of 3.84 µM. In contrast, P-Thr696-MYPT1^690−701^ had no influence on the activity PP2Ac. P-Thr696-MYPT1^690−701^ also inhibited Flag-MYPT1-PP1c, overexpressed and isolated from tsa201 cells [32] with an IC_50_ of 12.16 µM (Figure 2A) slightly higher than that of PP1c. The interaction between PP1c and P-Thr696-MYPT1^690−701^ was quantified using microscale thermophoresis (MST) and isothermal titration calorimetry (ITC) measurements (Figure 2B,C). MST provided a K_D_ of 33.21 ± 5.12 µM for the formation of the PP1c-P-Thr696-MYPT1^690−701^ complex (Figure 2B), while with ITC a K_D_ of 0.39 ± 0.18 µM was determined (Figure 2C). An attempt was also made to quantify the interaction of PP1c with MYPT1^690−701^, but in these cases, no MST or ITC signals suitable for quantitative evaluation were obtained. These data confirm a reasonable stable association between PP1c and P-Thr696-MYPT1^690−701^; however, the K_D_ values determined by the two methods for the interaction differ significantly. The reasons for this discrepancy are not yet clear.

### 2.2. Interaction of P-Thr696-MYPT1^690−701^ with PP1c Revealed by Saturation Transfer Difference NMR Measurements

To further characterize the PP1c-P-Thr696-MYPT1^690−701^ interaction saturation transfer difference (STD) [33,34], NMR measurements were performed. This technique is suitable for the identification of amino acid residues of P-Thr696-MYPT1^690−701^, which are involved in the interaction with PP1c. ^1^H-NMR spectra of P-Thr696-MYPT1^690−701^ was recorded in the absence of recombinant PP1cα (Figure 3A, lower panel) identifying the proton resonance signals associated with each amino acid of the peptide (see the sequence on Figure 3A, upper panel). Figure 3A (middle panel) illustrates the STD NMR spectrum of P-Thr696-MYPT1^690−701^ obtained in the presence of recombinant PP1cα. Strong STD signals were detected with the hydrophobic residues (Val699, Leu701) as well as Thr700, while the basic Arg and Gln residues provided weaker signals. STD ^1^H-NMR spectra of P-Thr696-MYPT1^690−701^ (1.3 mM) also recorded in the absence recombinant PP1c (Appendix A). The lack of signals in the control STD spectrum (Appendix A) confirms that selective irradiation applied at −650 Hz (−1.3 ppm) has no partial saturation effect on the P-T696-MYPT1^690−701^ signal intensities. Thus, the STD signals in spectrum (Appendix A) are solely due to saturation transferred from the protein upon binding interaction. Due to the severe overlap of proton resonances (Figure 3A) and the relatively poor signal-to-noise ratio (S/N) of the STD spectrum, no quantitative assessment of STD intensities was attempted. Thus, the STD signals were used only for qualitative epitope mapping. 

These data confirmed that the hydrophobic and basic regions of P-Thr696-MYPT1^690−701^ interacted with PP1c suggesting a “substrate-like” binding to the hydrophobic and acidic grooves of the enzyme as a major cause of the inhibition.

The dephosphorylation of P-Thr696-MYPT1^690−701^ was followed by real-time time-dependent ^1^H-NMR measurement. A gradual decrease in the signal intensity of the methyl protons (H_γ_) of P-Thr696 with a parallel increase in the corresponding signal of Thr700 was observed in the ^1^H-NMR spectrum as a function of time (Figure 3B, left), indicating dephosphorylation of the peptide during the measurement. The signal intensity of the Thr700 methyl protons was increased because the methyl proton resonance signal of Thr696 in the dephosphorylated peptide appeared at the same ^1^H chemical shift than that of Thr700. Plotting the integrated signal intensity of P-Thr696 as a function of time (Figure 3B, right) the half-life (t_1/2_ = 82 min) of the dephosphorylation of P-Thr696-MYPT1^690−701^ was determined indicating a rather slow dephosphorylation of the phosphopeptide by PP1c even at the same high concentration of the enzyme that used for the STD measurement.

### 2.3. Dephosphorylation of P-Thr696-MYPT1^690−701^, ^32^P-MLC20 and Their Combination by PP1c

To further confirm that P-Thr-MYPT1^690−701^ is a substrate for PP1c phosphatase assays were carried out at relatively high PP1c (100 nM) and P-Thr696-MYPT1^690−701^ (100 µM) concentrations to mimic similar conditions regarding the enzyme/substrate ratio as in the NMR analysis. The phosphate (P_i_) released from the peptide was determined by malachite green reagent [35,36]. Figure 4A illustrates that the dephosphorylation of P-Thr696-MYPT1^690−701^ proceeded slowly under these conditions and a half-life for the dephosphorylation (t_1/2_ = 87.9 min) extrapolated from the measured data was apparently similar to the value obtained from the NMR experiment. It was also of interest that how the presence of a phosphosubstrate (P-MLC20) may influence the dephosphorylation of the P-Thr696-MYPT1^690−701^ inhibitory peptide. In these experiments, the dephosphorylation of the P-Thr-MYPT1^690−701^ was determined by the decreased immunoreactivity of the phosphopeptide with anti-MYPT1^pThr696^ antibody on dot blots (Figure 4B) in the presence of 10 µM P-MLC20. It is seen that P-Thr696-MYPT1^690−701^ was dephosphorylated by PP1c slightly slower in the presence of P-MLC20 (t_1/2_ ~ 103.4 min) compared to that of its absence (Figure 4A, t_1/2_ ~ 87.9 min), presumably due to the mutual competition between the P-Thr696-MYPT1^690−701^ peptide and P-MLC20 substrate for binding at the substrate binding grooves at the catalytic site of PP1c. This presumed competition was further confirmed when the effect of P-Thr696-MYPT1^690−701^ was probed on the dephosphorylation of ^32^P-MLC20 (Figure 3C). The upper part of Figure 4C illustrates similarities in the amino acid sequences surrounding the phosphoserine and phosphothreonine residues in ^32^P-MLC20 and P-Thr696-MYPT1^690−701^, respectively. In the absence of P-Thr696-MYPT1^690−701 32^P-MLC20 was dephosphorylated with a half-life of 101.6 ± 8.7 s (~1.7 min) while in the presence of 10, 100 and 500 µM P-Thr696-MYPT1^690−701^ the half-life of the reaction increased to 149.7 ± 56.7 (~2.5 min), 227.9 ± 5.8 (~3.8 min) and 604.06 ± 0.02 (~10 min) seconds (Figure 4C). These results indicate that PP1c dephosphorylates ^32^P-MLC20 much faster than P-Thr696-MYPT1^690−701^; however, the latter competes with the substrates resulting in significant inhibition of ^32^P-MLC20 dephosphorylation reflected in the increasing half-life of the substrate. These data are in accordance with an “unfair competition” mechanism described previously for the inhibition of PP1 by phospho-CPI-17 [13].

### 2.4. PP1c-Peptide Docking

Multiple docking runs were performed with the P-Thr696-MYPT1 peptide or in which the phosphorylated threonine was replaced by phosphoserine (P-Ser696-MYPT1) (Figure 5). The lowest energy PP1c-peptide complexes were selected based on the best HADDOCK scores. The HADDOCK scores and the energies calculated are shown for the poses of peptides in the structure of the PP1c-peptide complexes in Appendix A. For the scoring, the software uses the OPLS force field (electrostatics and van der Waals energies), with additional empirical terms. Because of the different terms in the scoring function, HADDOCK uses arbitrary units, which are not suitable to predict binding affinity, only to compare different solutions for given complexes [37]. Based on the scores (Appendix A) the pattern of stabilization forces are similar for the P-Thr696-MYPT1-PP1c and the P-Ser696-MYPT1-PP1c complexes. 

It was apparent that the docking poses for P-Thr696-MYPT1 (Figure 5A) and P-Ser696-MYPT1 (Figure 5B) differed. In accord with the STD NMR results residues in the hydrophobic tail (Leu701, Thr700, and Val699) of both peptides docked to the hydrophobic groove of PP1c (Figure 5C,D). The P-Thr696-MYPT1 peptide (Figure 5C) contacted Arg96, Tyr134, Ile133, Arg132, Ala128, Asn131, Ile130 and Gly222, while the P-Ser696-MYPT1 peptide (Figure 5D) interacted with Tyr134, Ile133, Arg132, Asp194, Val195, Pro196, Ala128, Ser129 and Gly222 residues via van der Waals interactions, indicating quite similar binding fashion of the two peptides in the hydrophobic groove. The basic amino acid residues docked to the canonical substrate binding acidic groove of PP1c in case of P-Ser696-MYPT1 (Figure 5F). However, the docking pattern appeared to be different for P-Thr-MYPT1 with regards of the basic residues as they were docked to another acidic grove (Figure 5E) distinct from the substrate binding one. In this pose P-Thr696-MYPT1 managed to interact with PP1c forming several H-bonds (Figure 5E) with residues of Asp220, Arg221, Asn219, Glu218, Gln198, Gly217, Thr226, Asp208, Val213, Asp210. In contrast, a different interaction profile in case of P-Ser696-MYPT1 (Figure 5F) was identified where binding of the basic residues to Cys273, Glu218, Asp277 and Glu252 in PP1c were apparent.

The distances between the metal ions and the coordinating amino acid residues at the catalytic center were estimated in the docked complexes and compared to the ones derived from the crystal structure of PP1c with tungstate [6]. The data obtained from the docked structures and the crystal structure (see Table 1) were quite similar and there were only subtle differences except for a 0.9 Å difference in the distance of His173 to Mn_1_^2+^ in P-Thr-MYPT1^690−701^ (2.7 Å) and P-Ser696-MYPT1^690−701^ (1.8 Å).

The binding distances of other amino acid residues with suggested roles in catalysis [6,7], however, showed more pronounced differences between the PP1c-P-Thr696-MYPT1^690−701^ and PP1c-P-Ser696-MYPT1^690−701^ complexes (Figure 5G,H). Residues Arg96, Tyr272, His125 and Arg221 were closer to the phosphoryl residue in PP1c-P-Ser696-MYPT1^690−701^ (Figure 5H) than in PP1c-P-Thr696-MYPT1^690−701^ (Figure 5G), represented by the following differences of distances: Tyr272, 5.7 Å vs. 15.5 Å, His125, 4.5 Å vs. 11.1 Å, Arg221, 4.0 Å vs. 5.5 Å, respectively. There were only subtle differences in the distances of Asn124 (3.2 Å vs. 3.4 Å), and Arg96 (11.0 Å vs. 12.0 Å) to the phosphosubstrates in the two complexes. The distinct coordination pattern at the catalytic center and the stronger binding profile at a novel acidic groove of P-Thr696-MYPT1^690−701^ on the surface of PP1c might cause the slower dephosphorylation and dissociation of the peptide from PP1c causing inhibition of the binding and dephosphorylation of the phosphoserine substrate.

## 3. Discussion

In recent years, finding molecules or peptides to inhibit or activate PP1 has been regarded to possess important pharmacological relevancies as influencing the cellular activity of PP1 is a major regulatory factor in numerous signaling processes under normal or diseased conditions. Our present studies establish that a phosphorylated peptide (R^690^QSRRS(pT696)QGVTL^701^) from the MYPT1 regulatory subunit of MP is an effective and selective inhibitor of both PP1c and the MP holoenzyme (PP1c-MYPT1) with IC_50_ values (3–12 µM) at the lower micromolar range. These data are in accordance with previous results showing that the C-terminal fragment of a recombinant His-tagged MYPT1^514−963^ phosphorylated at Thr654 (corresponding to Thr696 in the longer MYPT1 isoform) inhibited PP1c with an IC_50_ of about 100 nM [30]. In contrast, GST-MYPT1^654−880^ fragment phosphorylated at Thr696 inhibited MP holoenzyme potently in nanomolar concentrations but were without effect on PP1c [31]. The above discrepancies might be due to the differently tagged protein fragments. Our present results confirm that a short peptide representing the inhibitory regions of phosphorylated MYPT1 (without any tags) still remains inhibitory for PP1c. P-Thr696-MYPT1^690−701^ exhibits decreased inhibitory effectiveness compared to His-tagged MYPT1^514−963^ or GST-MYPT1^654−880^, indicating that in the longer fragments presumably other regions beside the inhibitory sequence may also contribute to the stabilization of the interactions. Nevertheless, the shorter peptide, even with moderate inhibitory effectiveness, may be a good candidate as a template to design intracellular peptide inhibitors for PP1.

The P-Thr696-MYPT1^690−701^ peptide forms strong interaction with PP1c as assessed using MST (K_D_ = 33.2 µM) and ITC (0.39 µM) measurements although the dissociation constant determined by the two methods are quite different. The reasons for these discrepancies are not fully understood; however, it might be due to the fact that during the measurements targeting the interaction of PP1c with P-Thr696-MYPT1^690−701^, the phosphopeptide is dephosphorylated at least partially; thus, the combined effect of the interaction and the dephosphorylation processes are determined. The dephosphorylation of P-Thr696-MYPT1^690−701^ peptide by PP1c is established by ^1^H NMR experiments as well as in phosphatase assays with dephosphorylation half-life of t_1/2_ = 82–103 min compared to that of t_1/2_~1.7 min for P-MLC20 substrate (see Figure 4). These experiments are in accordance with the conclusion that the dephosphorylation of the inhibitory peptide is also catalyzed by PP1c, although with significantly slower rate than the natural substrate, P-MLC20. These data imply that the P-Thr696-MYPT1^690−701^ peptide inhibits PP1c by a so called “unfair competition with the substrate”, a mechanism established earlier for the inhibition of PP1c by CPI-17 against the P-MLC20 substrate [13].

Determination of the structure of PP1c-P-Thr696-MYPT1^690−701^ complex is complicated by the fact that dephosphorylation of the peptide occurs in time during the period of investigation. Therefore, docking simulations for the possible structures for PP1c with the inhibitory peptide has been completed to clarify the hypothetical structure of PP1c-P-Thr696-MYPT1^690−701^ complex. To understand the possible differences caused by the phosphorylated residues, simulations with both P-Thr696-MYPT1^690−701^ and P-Ser696-MYPT1^690−701^ were carried out (Figure 5). These molecular models obtained by docking simulations were similar with respect to binding the peptide in the hydrophobic groove in approximately the same fashion; however, the structure predicted at the catalytic center and the binding patterns of the basic sequences were distinct. First, the basic sequences of the P-Ser696-MYPT1^690−701^ peptide bound to the canonical acidic groove of PP1c while the same sequence of P-Thr696-MYPT1^690−701^ occupied another acidic region in which the interaction with PP1c was stabilized with numerous hydrogen bonds. Second, the coordination distances of the phosphoresidues of the P-Ser or P-Thr peptides with the PP1c side chains were also distinct, as Tyr272, His125 and Arg221 were farther away from the phosphate and the phosphoester bond in P-Thr696-MYPT1 compared to P-Ser696-MYPT1. Among these residues, His125 has special importance in the catalysis of dephosphorylation, as it serves as an acid in protonation of the leaving group oxygen atom, thereby accelerating the hydrolysis process [6,7]. The above data, therefore, are compatible with a mechanism in which the P-Thr696-MYPT1^690−701^ inhibitory peptide presumably is dephosphorylated at much slower rate than the P-Ser peptide (or the P-Ser containing P-MLC20) and dissociation of the dephosphorylated peptide from the enzyme is also hampered by the relatively strong interaction via its basic sequences.

These compact structural features of the interaction between P-Thr696-MYPT1^690−701^ and PP1c are also supported by the relatively high enthalpy and a negative entropy change of the interaction determined by ITC (Figure 2C). These results are also consistent with the conclusion that PP1c prefers P-Ser over P-Thr in protein or peptide substrates. This view is supported by the facts that all protein inhibitors (inhibitor-1, DARPP-32, CPI-17) of PP1c includes P-Thr at the inhibitory sites and they are presumably dephosphorylated slowly hindering the binding of the physiological substrates. Consistent with these views, the well-known physiological substrates of PP1 (e.g., glycogen phosphorylase, P-MLC20) are P-Ser proteins. The above data appear to support the preference of P-Ser over P-Thr in dephosphorylation of substrates by PP1. However, a recent study has proven that both PP1 and PP2A prefer P-Thr over P-Ser in dephosphorylation of a large set of P-Thr- and P-Ser-containing peptides [38]. Nevertheless, in case of the model system assayed in our present study comparison of the P-Thr-phosphorylated MYPT1 inhibitory peptide with the P-Ser-phosphorylated MLC20 substrate with similar sequences in both peptides around the phosphoresidues, a P-Ser preference over P-Thr appears to be obvious.

## 4. Materials and Methods

### 4.1. Materials

Reagents were purchased from the following sources: MYPT1^690−701^ and P-Thr696-MYPT1^690−701^ peptides were from Pepmic (Gusu District, Suzhou, China); bovine serum albumin (BSA), malachite oxalate, (NH_4_)_2_MoO_4_ and horseradish-peroxidase conjugated anti-rabbit antibody, anti-Flag-M2 Affinity-Matrix and Flag-peptide from Sigma-Aldrich (St. Louis, MO, USA); the potassium–phosphate standard was from Promega (Madison, WI, USA); Ser/Thr phosphatase assay kit, anti-MYPT1^pT696^ antibody from Millipore (MilliporeSigma, Burlington, MA, USA); Microcystin-LR was a kind gift from Dr. Csaba Máthé (Department of Botany, University of Debrecen). 

All other reagents and materials were of the highest purity available from commercial sources.

### 4.2. Proteins

The native catalytic subunits of protein phosphatase-1 and -2A (PP1c and PP2Ac) were purified from rabbit skeletal muscle [18]. The 20 kDa light chain of smooth muscle myosin (MLC20) was purified from turkey gizzard and phosphorylated as described previously [39]. Recombinant PP1cα [40] and PP1cδ [18] were expressed in *E. coli* and purified as described in the respective reference. Flag-MYPT1-PP1c was obtained by overexpressing Flag-MYPT1 in tsa201 cells [32], then immobilizing the Flag-MYPT1-PP1c complex on anti-Flag-M2 Affinity-Matrix. After extensive washing of the gel matrix 3 times with Tris-buferred saline (TBS), the Flag-MYPT1-PP1c was eluted by Flag-peptide in TBS.

### 4.3. Phosphatase Assays

Skeletal muscle PP1c (1.5–2.2 nM), PP2Ac (1.5 nM) or Flag-MYPT1-PP1c (2.5 nM) was assayed with 1 µM ^32^P-labelled MLC20 (^32^P-MLC20) in the presence or absence of nonphosphorylated (MYPT1^690−701^) or phosphorylated (P-Thr696-MYPT1^690−701^) MYPT1 peptide in 20 mM Tris/HCl (pH 7.4), 0.1% 2-mercaptoethanol at 30 °C. The ^32^P_i_ released from ^32^P-MLC20 substrate were determined as previously described [19]. The assay (80 µL) to determine the half-life of dephoshorylation for ^32^P-MLC20 (10 µM) in the presence of various concentration of P-Thr696-MYPT1^690−701^ (10–500 µM) was carried out in a similar fashion except that 5 µL aliquots were removed from the reaction mixture at different intervals for radioactivity measurements. 

### 4.4. Microscale Thermophoresis (MST) Experiments

The interaction of P-Thr696-MYPT1^690−701^ and the recombinant PP1cα (rPP1cα) was studied by Monolith^TM^ NT.LabelFree (MST power: 40%; LED Power: 10%) instrument (NanoTemper Technologies GmbH, Munich, Germany). A 16-point two-fold P-Thr696-MYPT1^690−701^ peptide dilution series was generated (9.16 nM-300 µM) in 25 mM Tris/HCl, 100 mM NaCl, 1 mM DTT, 0.1% PEG 8000, pH 8.0 and mixed with rPP1cα (2 µM). The mixtures were loaded into NT.LabelFree standard capillaries. The capillaries on the capillary tray were scanned and intrinsic fluorescence change of rPP1cα upon ligand binding was recorded. The data were analyzed by MO.Affinity Analysis v2.3 software.

### 4.5. Isothermal Titration Calorimetry (ITC)

Isothermal titration calorimetry (ITC) experiments were carried out with MicroCal ITC200 instrument. rPP1cα was dialyzed in 25 mM Tris/HCl (pH 8.0), 150 mM NaCl, 1 mM TCEP (assay buffer) for 3 h first, then 16 h with a fresh batch of solution. Prior to measurements, rPP1cα (0.5–2 µM) was transferred into the sample cell of the calorimeter while the reference cell was filled with the assay buffer. P-Thr696-MYPT1^690−701^ (5–20 µM) in the assay was applied to the syringe and 2 µL aliquot (19 times) was injected into the cell at a stirring rate of 300 rpm at 25 °C. The reference power was 5 µcal/s. The titration data were analyzed with MicroCal ITC-Origin software using one set of site model to obtain binding parameters of protein–ligand interaction.

### 4.6. Saturation Transfer Difference NMR

NMR experiments were performed with a Bruker Avance II 500 spectrometer operating at 500 MHz ^1^H frequency and equipped with a 5 mm z-gradient TXI probe head and carried out as previously described [18]. Both ^1^H-NMR and STD spectra were recorded with watergate-type residual water suppression scheme at 293 K on samples containing 1.3–1.5 mM P-(Thr696)MYPT1^690−701^ dissolved in 500 µL D_2_O containing 10 mM imidazole (pH = 6.5) and 150 mM NaCl. The samples for STD measurements contained 13.3–14.4 µM rPP1cα or rPP1cδ (14–15 µM, 600 µL) purified from *E. coli* expression and dialyzed against 100 mL of D_2_O containing 10 mM imidazole (pH = 6.5) and 150 mM NaCl with two changes of the dialyzing solution. The ^1^H STD spectra were recorded with 3 s semi-selective irradiation of the PP1c up-field resonances at −650 Hz (−1.3 ppm) by a train of Eburp 90° pulses of 50 ms each. In the reference experiment, the off-resonance frequency was set to −13,500 Hz (−27.0 ppm). The duration of the hard 90° proton pulse was 10.25 µs. Classical 2D NMR (^1^H-^1^H COSY, TOCSY, ROESY and ^1^H-^13^C HSQC) spectra were also recorded to achieve unambiguous resonance assignment of P-Thr696-MYPT1^690−701^. All spectra were processed and analyzed with Topspin 2.1 software of Bruker Biospin.

### 4.7. Malachite Green Assay

The malachite green reagent was generated as follows: malachite oxalate and (NH_4_)_2_MoO_4_ were dissolved in 1.2 M HCl solution with the final concentrations of 2 mM and 100 mM, respectively. The solutions were mixed in 1:1 ratio and swirled for 20 min then filtered with Filtropure S 0.45. Tween 20 was added to the reagent (0.08%(*V*/*V*)) prior to use [35,36]. The reagent is stable for 3 weeks in the absence of detergent. The dephosphorylation of P-Thr696-MYPT1^690−701^ by PP1c was assessed using the malachite green assay to measure the P_i_ released from the peptide. A quantity of 100 nM nPP1c was preincubated in 20 mM Tris/HCl (pH 7.4), 0.02% 2-mecaptoethanol and 0.1% TritonX-100 (assay buffer) for 1 min at 30 °C then the reaction was initiated by the addition of P-Thr696-MYPT1^690−701^ (100 µM). Samples (15 µL) were removed from the reaction mixture 0.33, 0.66, 1, 2, 5, 10, 20, 30, 60 min and mixed with 3.4 M HCl (3 µL) solution. Control samples representing the initial conditions were generated by adding all the components into 3.4 M HCl solution. The malachite green assay was performed on a 384-well plate. The samples (18 µL) were complemented to 30 µL with assay buffer, then added to 30 µL malachite green reagent solution (1 mM malachite oxalate, 50 mM (NH_4_)_2_MoO_4_ and 0.08% (*V*/*V*) Tween-20 in 1.2 M HCl solution) on the plate. After 20 min incubation, the absorbance of the samples at 630 nm due to the color intensity changes of the liberated P_i_ from the substrate was determined by Thermo Multiscan go plate reader instrument. The liberated P_i_ contents (pmol) were subtracted from P-Thr696-MYPT1^690−701^ and the residual values plotted as the function of time. A calibration curve with potassium phosphate standard (250 to 1500 pmol range) was also established for the exact determination of the released P_i_.

### 4.8. Dot Blot Assay

The dephosphorylation of 100 µM P-Thr696-MYPT1^690−701^ by PP1c (100 nM) in the presence of 10 µM P-MLC20 was followed using a dot blot assay. PP1c was mixed with P-MLC20 at 30 °C in 20 mM Tris/HCl (pH 7.4), 0.1% 2-mercaptoethanol and then P-Thr696-MYPT1^690−701^ was added. Aliquots were removed at different intervals, mixed with 1 µM Microcystin-LR (1 µL) to inactivate PP1c; then, these samples were dripped to nitrocellulose membrane. The membrane was blocked with 5% BSA in Tris-buffered saline plus Tween-20 (TBST) at room temperature for 2.5 h. Then membrane was incubated with rabbit anti-MYPT1^pT696^ antibody in (1000-fold dilution in 5% BSA in TBST) for 16 h at 4 °C. After washing the membrane was incubated with horseradish–peroxidase-conjugated anti-rabbit antibody (7000-fold dilution in 5% BSA in TBST) for 2 h at room temperature; then, after washing, the immunoreactions were developed with enhanced chemiluminescence reagent, recorded with BIO-RAD ChemiDock^TM^ Touch imaging system and the data were analyzed with ImageJ 1.53c software.

### 4.9. Peptide Docking

For molecular docking the structure of PP1cγ isoform bound to okadaic acid was used (PDB ID: 1JK7; [8]) as a starting point. For peptide docking analysis and to determine the best overall structure of the P-Thr696-MYPT1^690−701^-PP1γ complex, HADDOCK 2.4 web portal was used [37,41]. The docking was carried out as a standard protein–peptide docking according to the web portal. The docking interface (active amino acids in the interaction) was defined by selecting residues in a 10 Å radius from the okadaic acid bound at the catalytic center of PP1γ. These residues were defined as fully flexible. The His125 residue was fully protonated according to a proposed reaction mechanism [6], the rest of the histidine residue protonation states were determined by HADDOCK. We defined the whole peptide (12 residues) as active. The peptides were fully flexible during docking, enabling the software to find the best overall pose for the peptide. For docking, distance restraints (surface contact restraint, and center of mass restraint) were used to maintain proximity between the peptides and PP1c. Every other parameter remained as default inside HADDOCK, except for the analysis mode, which was set to full. Molecular graphics, distance measurements, H-bond, and van der Waals contact analyses were performed with UCSF ChimeraX software [42].

## 5. Conclusions

Our present results imply, based on in vitro assays, that P-Thr696-MYPT1^690−701^ peptide may serve as a good pharmacophore model for designing peptide inhibitors of PP1 for possible cellular application. The advantage of P-Thr696-MYPT1^690−701^ being used as a cellular PP1 inhibitor is that this peptide inhibits not only PP1c, but PP1 holoenzyme, too. These data imply that the peptide does not compete with the regulatory proteins for binding to PP1c. However, the peptide may need modification to be able to permeate cell membranes to apply as an intracellular inhibitor of PP1. The latter may be achieved by adding a few more basic residues at the N-terminus of the peptide to the existing basic sequence creating a basic stretch reminiscent of the HIV–TAT membrane-penetrating sequence [43]. In addition, retro-inverso peptide could be synthesized to avoid proteolytic degradation [44], increasing the intracellular lifetime of the inhibitory peptide. These experiments may provide further perspectives in examining the possibilities for application of peptides in pharmacological influences of P-Ser/Thr-specific protein phosphatases.

## Figures and Tables

**Figure 1 ijms-24-04789-f001:**
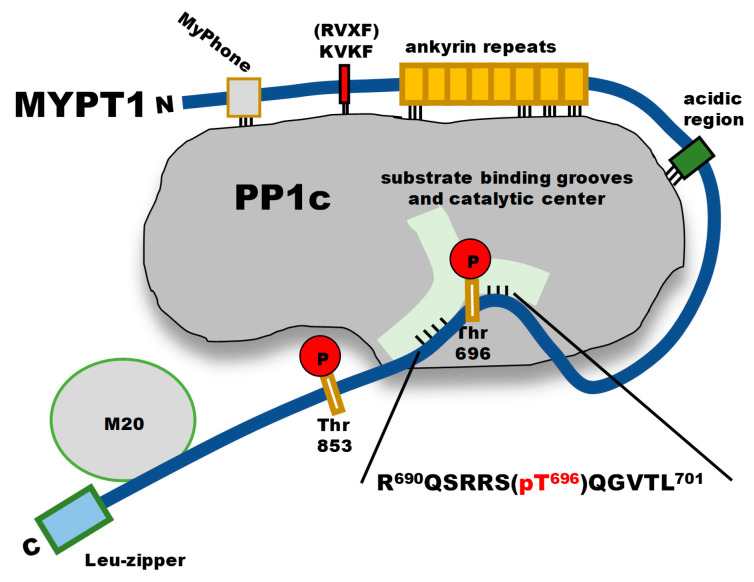
The structure of myosin phosphatase holoenzyme depicting interactions between PP1c, phosphorylated MYPT1 and M20 subunits.

**Figure 2 ijms-24-04789-f002:**
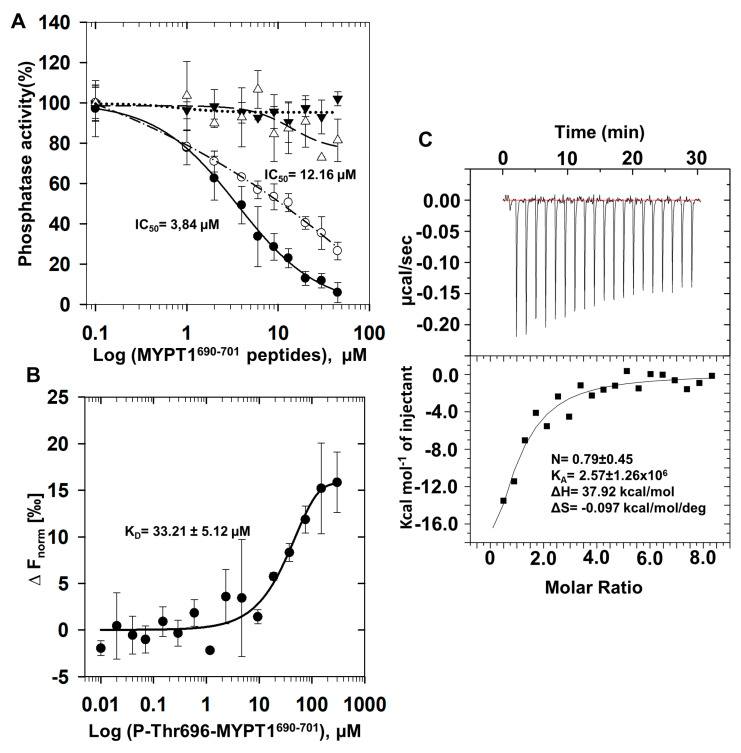
Effects of a phosphorylated peptide from MYPT1 on the activity of PP1c, PP2Ac and myosin phosphatase holoenzyme (Flag-MYPT1-PP1c). (**A**): The native catalytic subunit of PP1 (PP1c; ▼,●) or PP2A (PP2Ac; ∆) purified from rabbit skeletal muscle or Flag-MYPT1-PP1c overexpressed and isolated from tsa201 cells (○) was assayed in the presence of dephosphorylated (▼) or phosphorylated (○,●,∆) MYPT1 peptide of R^690^QSRRS(pT696)QGVTL^701^ (P-Thr696-MYPT1^690−701^) with ^32^P-MLC20 substrate as described in Section 4. Values are mean ± SD (n = 4). Phosphatase activity in the absence of peptide was taken 100%. (**B**): Interaction of recombinant PP1cα with the phosphorylated peptide assessed by microscale thermophoresis (MST) or (**C**): with isotherm titration calorimetry (ITC) as described in Section 4.

**Figure 3 ijms-24-04789-f003:**
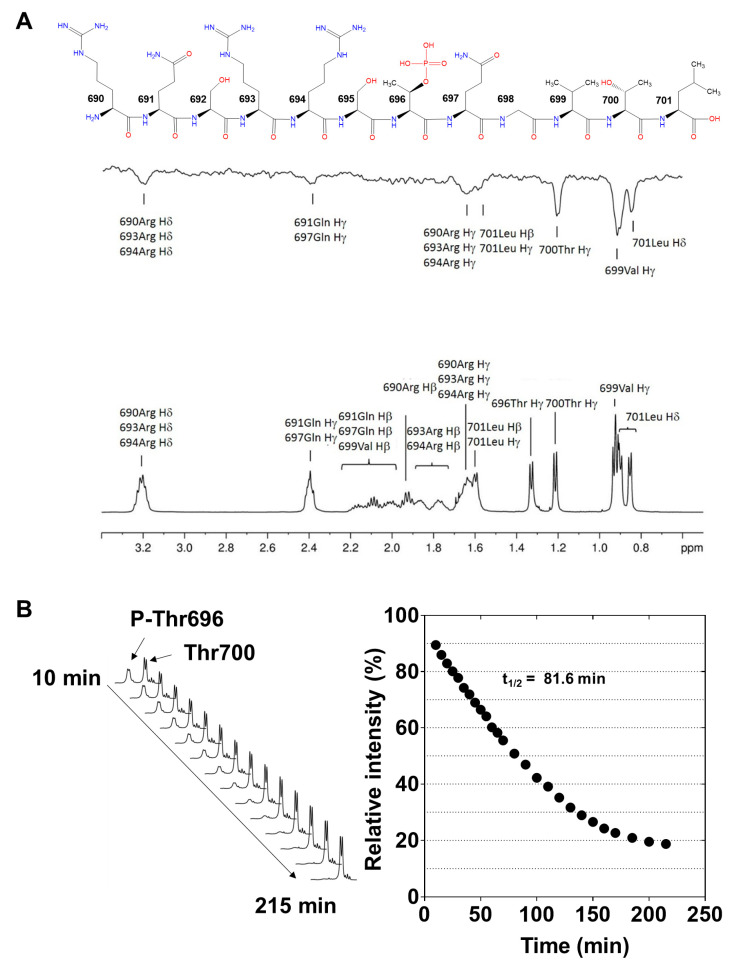
NMR spectroscopic investigations on the binding of P-Thr696-MYPT1^690−701^ to PP1c. (**A**): The structure of P-Thr696-MYPT1^690−701^ peptide is depicted (**upper panel**) and its ^1^H-NMR spectrum in D_2_O is shown (**lower panel**). Saturation transfer difference (STD) ^1^H-NMR spectrum of P-Thr696-MYPT1^690−701^ (1.3 mM) in the presence of 13.3 µM recombinant PP1c (**middle panel**). PP1cα and PP1cδ expressed in *E. coli* and purified from the bacterial lysate were used with similar results. The representative data presented here were obtained with PP1cα and P-Thr696-MYPT1^690−701^. (**B**): Dephosphorylation of P-Thr696-MYPT1^690−701^ followed by real-time ^1^H-NMR experiment. **Left**: changes of the signal intensity of P-Thr696 and Thr700 as a function of time. **Right**: the signal intensity of P-Thr696 plotted against the time allows the determination of the half-life (t_1/2_) for the dephosphorylation of P-Thr696-MYPT1^690−701^.

**Figure 4 ijms-24-04789-f004:**
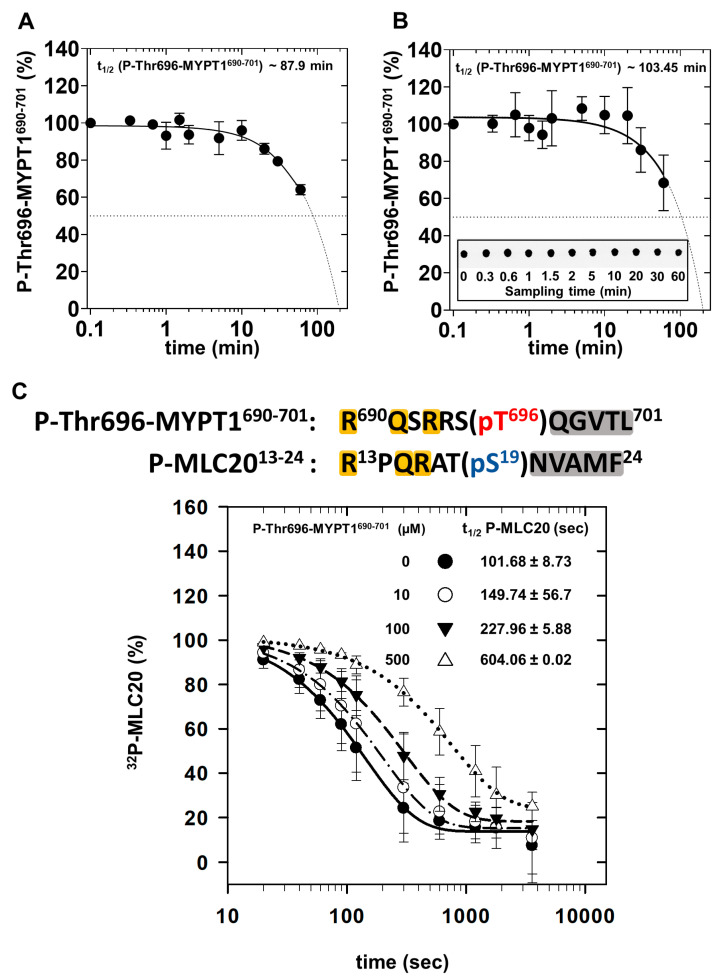
Dephosphorylation of P-Thr696-MYPT1^690−701^ and ^32^P-MLC20 by PP1c. (**A**): Phosphatase assays were carried out in the presence of 100 µM P-Thr696-MYPT1^690−701^ and 100 nM of rabbit skeletal muscle PP1c as described in Section 4. Dephosphorylation of the peptide was assessed by measuring the released P_i_ by malachite green assay. Data are mean ± SEM, n = 3 (three independent measurements with three parallels for each). (**B**): The dephosphorylation of 100 µM P-Thr696-MYPT1^690−701^ by 100 nM rabbit skeletal muscle PP1c was carried out in the presence of 10 µM P-MLC20 and the amount of P-Thr696-MYPT1^690−701^ was quantified on dot blots at different intervals using anti-MYPT1^pThr696^ antibody. Immunoreactions were detected using enhanced chemiluminescence, and signals were analyzed with ImageJ software as described in Section 4. Data are mean ± SD (n = 5). The mean values of data were fitted to the equations (sigmoidal curve, four-parameter Hill equation) using GraphPad Prism software and the curve was extrapolated to estimate the half-life of P-Thr696-MYPT1^690−701^ dephosphorylation. (**C**): Competition of P-Thr696-MYPT1^690−701^ with ^32^P-MLC20 substrate during dephosphorylation by PP1c. The peptide sequences of P-Thr696-MYPT1^690−701^ and ^32^P-MLC20 around the phosphorylation sites (pT696 and pS19) are aligned to emphasize similarities in the basic and hydrophobic stretches (orange and gray background, respectively) N-terminal and C-terminal to the phosphorylation sites. Rabbit skeletal muscle PP1c (100 nM) was preincubated in the absence or presence of P-Thr696-MYPT1^690−701^ at 30 °C for 1 min in the concentrations indicated in the inset of the figure; then, the reaction was started by the addition of 10 µM of ^32^P-MLC20. Aliquots were removed at different intervals and the released ^32^P_i_ was determined as described in Section 4. The released ^32^P_i_ was used to calculate the amount of ^32^P-MLC20 and the percent decrease in ^32^P-MLC20 was plotted to determine the half-life for dephosphorylation.

**Figure 5 ijms-24-04789-f005:**
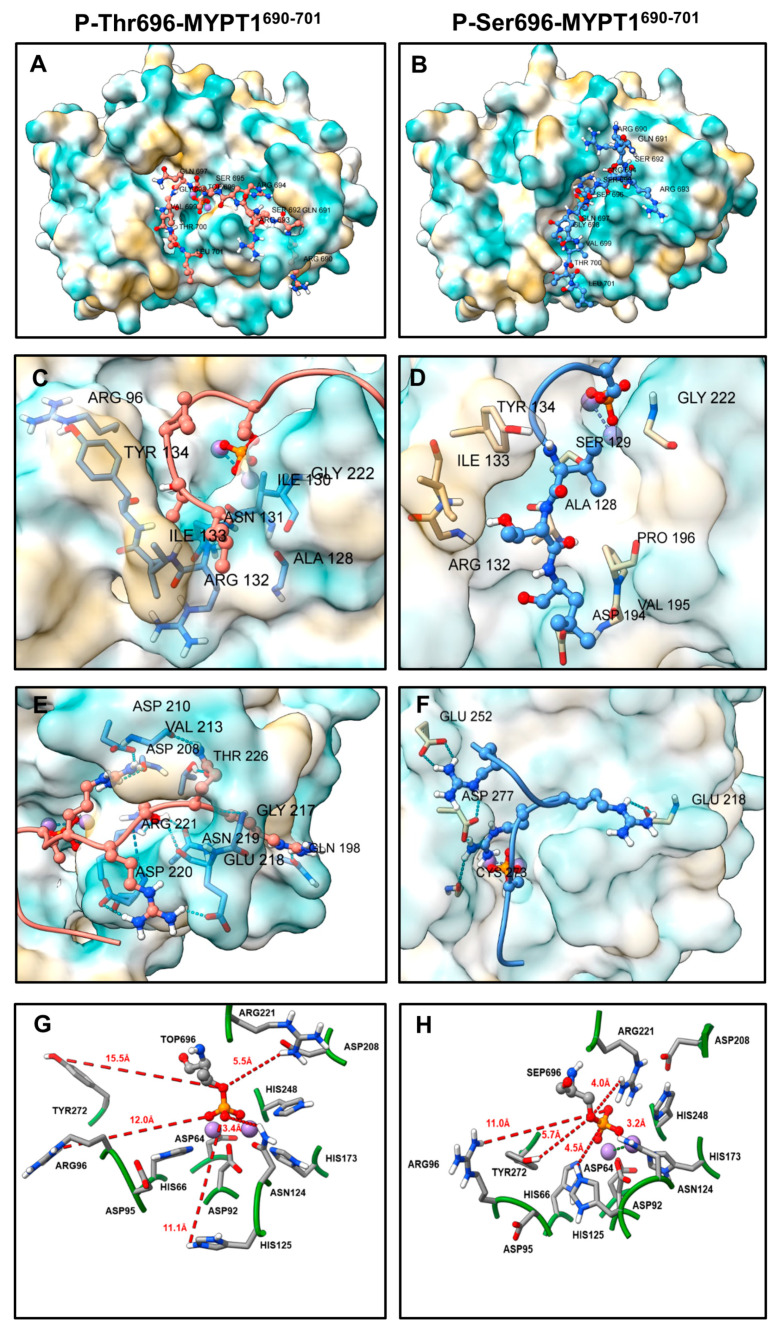
Poses of P-Thr696-MYPT1^690−701^ and P-Ser696-MYPT1^690−701^ peptide on the surface of PP1c derived from molecular docking simulations. Poses of P-Thr696-MYPT1^690−701^ (**A**) and P-Ser696-MYPT1^690−701^ (**B**) peptides on PP1c. Enlarged images of the binding of hydrophobic residues of P-Thr696-MYPT1^690−701^ (**C**) and P-Ser696-MYPT1^690−701^ (**D**) at the hydrophobic groove of PP1c. Enlarged images for the binding of basic residues of P-Thr696-MYPT1^690−701^ (**E**) and P-Ser696-MYPT1^690−701^ (**F**) to PP1c. Distances of the PP1c amino acid side chains from the coordinated phosphoryl residue of the substrate in the docked complexes of PP1c-P-Thr696-MYPT1^690−701^ (**G**) and PP1c-P-Ser696-MYPT1^690−701^ (**H**).

**Table 1 ijms-24-04789-t001:** Distances between the metal ions and their coordinating amino acid residues in docked PP1c–peptide complexes and in crystal structure.

		Distance (Å)		
Metal Ions *	Coordinating Residues **	PP1c Crystal	PP1c-P-Thr696-MYPT1^690−701^	PP1c-P-Ser696-MYPT1^690−701^
Mn_1_^2+^	Asp92	2.3	2.3	2.1
Mn_1_^2+^	Asn124	2.1	1.9	2.0
Mn_1_^2+^	His173	2.1	2.7	1.8
Mn_1_^2+^	His248	2.3	2.4	2.9
Mn_2_^2+^	Asp64	2.0	1.5	1.6
Mn_2_^2+^	His66	2.3	2.4	3.0
Mn_2_^2+^	Asp92	2.3	1.9	1.6

* Both of the metal ions are Mn^2+^ in the crystal structure of PP1cγ used for docking [8]. ** Distances between metal ions (Zn^2+^ and Fe^2+^) in PP1c crystal with tungstate and coordinating residues are from Egloff et al. [6].

## Data Availability

Data will be made available on request.

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
