# Peer review of "Phosphorylated Peptide Derived from the Myosin Phosphatase Target Subunit Is a Novel Inhibitor of Protein Phosphatase-1"

_ijms, 2023, doi:10.3390/ijms24054789_

Round 1

Reviewer 1 Report

The title is very long and unclear.

Abstract

It must be improved. The inhibitor or inhibitors and target enzyme should be clearly stated. Unfortunately, as the title is also unclear, at first reading is difficult to obtain an overview of this work.

Introduction

Further improvements are also required. I suggest that the authors elaborate on the description of the target enzyme properly. Myosin phosphatase is a complex enzyme composed of three subunits, one of which is the catalytic subunit, also called PP1 (protein phosphatase 1), the other subunit is the myosin binding subunit (MYPT1), and the third is one of unknown function (M20).

For example, in lines 89-90, it is confusing the statement: “Myosin phosphatase (MP) consists of PP1cd and myosin phosphatase target subunit-1 (MYTP1)" Instead must be re-written as “Myosin phosphatase (MP) consisting of a PP1cd subunit, a myosin binding subunit-1 (MYTP1), and unknown function subunit …"

A figure of the overall structure of the MP in the Introduction can help to better understand the description of different parts of the enzyme, including the RVxF motif mentioned in the Introduction.

The paragraph starting in line 120 presents the research question; however, it has not been clearly written. The statement regarding the origin of the peptide seems ambiguous.

Results and discussion

Lines 120–140 are no results. All of this text reference several literature sources. This section is better suited to the Introduction. Indeed, moving this text to the introduction section can help clarify the research question.

The dissociation constants (Kd) determined using microscale thermophoresis (33.2 μM), and isothermal titration calorimetry differed significatively (0.39 μM). In this situation, a third method must be used to clarify the discrepancies. In addition, molecular models and docking studies did not reveal binding energies.

The other results appear rigorous, and the figures are presented in good quality.

Material and methods

Writing skills can also be improved. In addition, scientific names must be written in italics, for example, for E. coli.

Conclusion

As the research question is not clearly stated and the results show significant differences between methods, it is difficult to support a conclusion.

Author Response

Responses to Reviewer 1

We would like to thank the Reviewer for her/his useful criticisms which have helped us to revise our manuscript.

Comment 1

The title is very long and unclear.

Response 1

The title was changed and simplified as „Phosphorylated peptide derived from the myosin phosphatase target subunit is a novel inhibitor of protein phosphatase-1” expressing better the subject of the manuscript.

Comment 2

Abstract: It must be improved. The inhibitor or inhibitors and target enzyme should be clearly stated. Unfortunately, as the title is also unclear, at first reading is difficult to obtain an overview of this work

Respone 1:

We respectfully note that inhibitor and target enzymes are clearly stated in the abstract as the second sentence says: „In this study we prove that a phosphorylated peptide of the inhibitory region of myosin phos-phatase (MP) target subunit (MYPT1), R690QSRRS(pT696)QGVTL701 (P-Thr-MYPT1690-701), interacts with and inhibits the PP1 catalytic subunit (PP1c, IC50=3.84 µM) and the MP holoenzyme” (Flag-MYPT1-PP1c, IC50=3.84 µM).

Comment 3

Introduction: Further improvements are also required. I suggest that the authors elaborate on the description of the target enzyme properly. Myosin phosphatase is a complex enzyme composed of three subunits, one of which is the catalytic subunit, also called PP1 (protein phosphatase 1), the other subunit is the myosin binding subunit (MYPT1), and the third is one of unknown function (M20). 
For example, in lines 89-90, it is confusing the statement: “Myosin phosphatase (MP) consists of PP1cd and myosin phosphatase target subunit-1 (MYTP1)" Instead must be re-written as “Myosin phosphatase (MP) consisting of a PP1cd subunit, a myosin binding subunit-1 (MYTP1), and unknown function subunit". A figure of the overall structure of the MP in the Introduction can help to better understand the description of different parts of the enzyme, including the RVxF motif mentioned in the Introduction.
The paragraph starting in line 120 presents the research question; however, it has not been clearly written. The statement regarding the origin of the peptide seems ambiguous.

Response 3

In the revised version of our manuscript we give a better description of the myosin phosphatase holoenzyme with mentioning the M20 subunit and discussing its binding in the holoenzyme. In addition, we present a figure depicting the subunit interactions of PP1c and the phosphorylated MYPT1 in the inactive myosin phosphatase holoenzyme. This figure also points to the inhibitory phosphorylated peptide from MYPT1, which is tested in this manuscript as a PP1 inhibitor, therefore, the origin of the peptide is now hopefully straightforward. The description of the research question and the brief account of the results achieved are also shorthened and simplified (see highlighted text and Figure 1 in Introduction).

Comments 4

Results and discussion: Lines 120–140 are no results. All of this text reference several literature sources. This section is better suited to the Introduction. Indeed, moving this text to the introduction section can help clarify the research question.

Response 4

The text in the objected lines was rearranged, partly omitted or partly included in the Introduction.

Comment 5

The dissociation constants (Kd) determined using microscale thermophoresis (33.2 μM), and isothermal titration calorimetry differed significatively (0.39 μM). In this situation, a third method must be used to clarify the discrepancies. In addition, molecular models and docking studies did not reveal binding energies.

Response 5

We note in the discussion that the exact determination of the interaction of PP1c with the phosphopeptide P-Thr696-MYPT1690-701 is influenced by the dephosphorylation of the peptide during the assays. We hypothesize that this process may shift the KD values oppositely from the real ones in case of MST and ITC measurements. The IC50 values (3-12 µM) for the inhibition of PP1c or PP1c-MYPT1 fall between the determined KD values. Although, IC50 values could not be considered as KDs, nevertheless they are indicators of enzyme-inhibitor interactions, thus partly confirming our hypothesis.   

Concerning binding energies by docking studies we note that the HADDOCK scores and the energies calculated are shown for the poses of peptides in the structure of the PP1c-peptide complexes in Table S1. Because of the different terms in the scoring function, HADDOCK uses arbitrary units, which are not suitable to predict binding affinity, only to compare different solutions for given complexes (see highlighted text 2.4. PP1c-peptide docking).

.

Comment 6

The other results appear rigorous, and the figures are presented in good quality.

Response 6

Thank you for these recognitions.

Comment 7

Material and methods. Writing skills can also be improved. In addition, scientific names must be written in italics, for example, for E. coli.

Response 7

Thank you for your note. They are corrected.

Comment 8

Conclusion: As the research question is not clearly stated and the results show significant differences between methods, it is difficult to support a conclusion.

Response 8

We hope that with the changes we have made in the revised manuscript the conclusions could be acceptable.

Reviewer 2 Report

The authors describe enzymatic binding and reactivity studies with protein phosphatase 1, specifically myosin phosphatase and subunits thereof. To this end, the binding of phospho-peptides to the PP1c subunit and to the holo-enzyme was investigated by iTC and MST experiments, and the inhibitory effect of these peptides was determined in de-phosphorylation assays. Furthermore, the binding epitope of P-MYPT1 (690-701) when interaction with PP1c was characterized by STD NMR spectroscopy, and a potential binding mode was determined by docking studies.

Overall, experiments have been performed with great care and are described in the methods section accordingly. The data presented is of high quality and the interpretation appears to be plausible. Discrepancies within the data, e.g. differing KD values of MST and iTC measurements, are clearly indicated, and apparent inconsistencies with recent findings in the literature are openly discussed. In conclusion, the manuscript appears well suited for publication in IJMS.

Nevertheless, I would like to raise the following points to improve the manucript:

- P(Ser)-MYPT1 should be named P(Ser696)-MYPT1 throughout the manuscript.

- STD NMR: A control experiment with the peptide in the absence of protein should be provided in the SI.

- STD NMR: STD effects should be quantified as %STD or STD amplification factor.

- STD NMR: How does P(Ser696)-MYPT1 (690-701) perform in STD NMR? Does the altered binding mode found in the docking studies lead to differences in the group epitope mapping?

- STD NMR: How was the buffer exchange/up concentration of the proteins achieved for sample preparation? Or, from what kind of stock solution were the proteins added?

- Docking poses shown in Fig. 4 should be provided as coordinate files in the SI.

- Moderate issues with English language.

Author Response

Responses to Reviewer 2

We would like to thank the Reviewer for the positive evaluation of our manuscript and for her/his useful notes which have helped us to improve the revised version.

Comment 1

P(Ser)-MYPT1 should be named P(Ser696)-MYPT1 throughout the manuscript.

Response 1

In the revised manuscript the term P-Thr696-MYPT1690-701 and P-Ser696-MYPT1690-701 are used throughout.

Comment 2

-STD NMR: A control experiment with the peptide in the absence of protein should be provided in the SI.

Response2

Thanks for this note. Now the result of the control STD experiment is included in the SI as Fig. S1. The lack of signals in the control STD spectrum (A) confirms that selective irradiation applied at -650 Hz (-1.3 ppm) has no partial saturation effect on the P-Thr696-MYPT1690-701 signal intensities. Thus, the STD signals in spectrum (B) and also in Fig. 3A of the manuscript are solely due to saturation transferred from the protein upon binding interaction. These statements are now included in the revised manuscript (see highlighted text in 2.2.).

Comment 3

STD NMR: STD effects should be quantified as %STD or STD amplification factor.

Response 3

Due to the severe overlap of proton resonances (Fig. 3A) and the relatively poor signal-to-noise ratio (S/N) of the STD spectrum no quantitative assessment of STD intensities was attempted. Thus, the STD signals were used only qualitatively for epitope mapping. These statements are now included in the revised manuscript (see highlighted text in 2.2.).

Comment 4

 STD NMR: How does P(Ser696)-MYPT1 (690-701) perform in STD NMR? Does the altered binding mode found in the docking studies lead to differences in the group epitope mapping?

Response 4

P(Ser696)-MYPT1 (690-701) was used only in docking studies as a phosphoserine alternate of the inhibitory phosphothreonine peptide. This phosphoserine peptide was not synthesized and used in experiments. We presume that P(Ser696)-MYPT1 (690-701) would be dephosphorylated much faster than P(Thr696)-MYPT1 (690-701) as it is exampled by the P-Ser-MLC20 with similar sequences around  the phosphoserine. Therefore, we assumed that the lifetime of the phosphoserine peptide would not be long enough for STD NMR study.

Comment 5

STD NMR: How was the buffer exchange/up concentration of the proteins achieved for sample preparation? Or, from what kind of stock solution were the proteins added?

Response 5

Thank you for directing our attention to this missing information. For buffer exchange, purified rPP1cα or rPP1cδ (14-15 µM, 600 µl) was dialysed against 100 ml of D2O containing 10 mM imidazole (pH = 6.5) and 150 mM NaCl with two changes of the dialyzing solution.

Comment 6

Docking poses shown in Fig. 4 should be provided as coordinate files in the SI.

Response 6

Coordinate files for the docking poses will be uploaded during the submission of the revised version.

Round 2

Reviewer 1 Report

The manuscript has significantly improved. I suggest the current version for publication.